# The Anti-Obesity Effect of Fish Oil in Diet-Induced Obese Mice Occurs via Both Decreased Food Intake and the Induction of Heat Production Genes in Brown but Not White Adipose Tissue

**DOI:** 10.3390/ijms26010302

**Published:** 2024-12-31

**Authors:** Takahiko Obo, Hiroshi Hashiguchi, Eriko Matsuda, Shigeru Kawade, Kazuma Ogiso, Haruki Iwai, Koji Ataka, Osamu Yasuda, Aiko Arimura, Takahisa Deguchi, Katsutaro Morino, Akihiro Asakawa, Yoshihiko Nishio

**Affiliations:** 1Department of Diabetes and Endocrine Medicine, Graduate School of Medicine and Dental Sciences, Kagoshima University, Kagoshima 890-8544, Japan; k3239763@kadai.jp (T.O.); hhashiguchi03171967@gmail.com (H.H.); k4962175@kadai.jp (S.K.); nomushi@kufm.kagoshima-u.ac.jp (K.O.); aiko-a@m.kufm.kagoshima-u.ac.jp (A.A.); degdeg@m3.kufm.kagoshima-u.ac.jp (T.D.); ynishio@m3.kufm.kagoshima-u.ac.jp (Y.N.); 2Department of Gene Therapy and Regenerative Medicine, Graduate School of Medicine and Dental Sciences, Kagoshima University, Kagoshima 890-8544, Japan; matsuda@m2.kufm.kagoshima-u.ac.jp; 3Department of Oral Anatomy and Cell Biology, Graduate School of Medicine and Dental Sciences, Kagoshima University, Kagoshima 890-8544, Japan; haruki@dent.kagoshima-u.ac.jp; 4Department of Health and Nutrition, Faculty of Nursing and Nutrition, Kagoshima Immaculate Heart University, Kagoshima 895-0011, Japan; kataka@k-jundai.jp; 5Department of Sports and Life Sciences, National Institute of Fitness and Sports in Kanoya, Kanoya 891-2311, Japan; o-yasuda@nifs-k.ac.jp; 6Department of Psychosomatic Internal Medicine, Graduate School of Medicine and Dental Sciences, Kagoshima University, Kagoshima 890-8544, Japan; asakawa@m2.kufm.kagoshima-u.ac.jp

**Keywords:** omega-3 polyunsaturated fatty acids, diet-induced obesity, brown adipose tissue, white adipose tissue

## Abstract

Omega-3 (ω-3) polyunsaturated fatty acids in fish oil have been shown to prevent diet-induced obesity in lean mice and to promote heat production in adipose tissue. However, the effects of fish oil on obese animals remain unclear. This study investigated the effects of fish oil in obese mice. C57BL/6J mice were fed a lard-based high-fat diet (LD) for 8 weeks and then assigned to either a fish oil-based high-fat diet (FOD) or continued the LD for additional 8 weeks. A control group was fed a standard diet for 16 weeks. Mice fed the FOD showed weight loss, reduced adipose tissue mass, and lower plasma insulin and leptin levels compared to those fed the LD. Rectal temperatures were higher in the FOD and LD groups compared to the control group. Energy intake was lower in the FOD group than the LD group but similar to the control group. The FOD and LD groups exhibited increased expression of heat-producing genes such as *Ppargc1a*, *Ucp1*, *Adrb3*, and *Ppara* in brown adipose tissue but not in white adipose tissue. The FOD reduced food consumption and increased rectal temperature and heat-producing genes in brown adipose tissue. Fish oil may therefore be a potential therapeutic approach to obesity.

## 1. Introduction

Since 1980, the prevalence of obesity has doubled in more than 70 countries and continues to increase in most other countries [1]. In 2024, the World Obesity Coalition warned that more than half of the world’s population would be classified as obese or overweight by 2035 [2]. Although some reports suggest that fish oil (FO) exerts a favorable impact on the treatment and prevention of obesity, heterogeneity of effects exists [3,4]. Conversely, numerous reports have directed the effects of FO against body weight gain in rodents [5,6]. The ω-3 polyunsaturated fatty acids contained in fish oil (FO) increase β_3_-adrenergic receptor (β_3_AR) expression in brown adipose tissue (BAT) and white adipose tissue (WAT) via sympathetic activation [6]. This partly explains the increased expression of uncoupling protein 1 (UCP1) and heat production in the BAT and WAT of nonobese mice [7,8,9]. Sympathetic stimulation of WAT causes the transformation of WAT into beige adipose tissue, which has a thermogenetic capacity similar to brown adipocytes [10,11,12]. Taken together, evidence suggests that FO increases the expression of UCP1 in both BAT and WAT [6], which causes heat production and reduces weight gain [7,13]. However, although there is evidence regarding FO stimulation of heat-producing genes such as *Ppargc1a* (peroxisome proliferator-activated receptor γ coactivator-1α), *Ucp1*, *Adrb3* (β_3_AR), and *Ppara* (peroxisome proliferator-activated receptor-α) in the BAT and WAT of nonobese mice [6,13,14], the effect of FO in obese mice remains unclear. The study looked at whether feeding FO to obese mice would also make them lose weight.

## 2. Results

### 2.1. Fish Oil Results in Body Weight Loss and Lowered Caloric Intake While Maintaining Elevated Rectal Temperatures in Mice with Diet-Induced Obesity

Five-week-old male C57BL/6J mice were fed a CD or an LD for 8 weeks. Mice fed an LD for 8 weeks were randomly divided into two groups: one group was maintained for another 8 weeks on the LD and the other group was introduced to a FOD for 8 weeks (Figure 1a). As shown in Figure 1b, at 12 weeks of age, the diet-induced obesity (DIO) model mice (LD-fed) weighed significantly more than the nonobese, CD-fed mice (33.4 ± 2.1 g vs. 27.2 ± 1.1 g, *p* < 0.001). During the experimental period, the body weight of mice fed the LD increased in a time-dependent manner (Figure 1b,c). At the end of the experimental period, mice fed the LD had gained significantly more weight than mice fed either the CD or the FOD (both *p* < 0.001; Figure 1c).

Consistent with body weight, WAT masses of the mesentery (Figure 2a), inguinal (Figure 2b), and epididymis regions (Figure 2c) were significantly lower in mice fed the FOD and the CD than in mice fed the LD (all *p* < 0.05). Significantly higher liver weights were observed in mice fed both the LD and the FOD compared with those fed the CD (both *p* < 0.001; Figure 2d). However, Oil Red O-stained images of liver tissue revealed that fat deposition was observed only in mice fed the LD (Figure 2e), indicating that the increase in liver weight observed in mice fed the FOD did not reflect fatty liver. Mice fed the LD had a significantly higher caloric intake per day than the other two groups (both *p* < 0.01; Figure 2f,g). Mice fed the FOD had a temporary decrease in caloric intake when switching from the LD to the FOD at 13 weeks, but thereafter, caloric intake was similar to that of mice fed the CD (Figure 2f,g). Rectal temperatures in mice fed the LD and the FOD were significantly higher than in those fed the CD (both *p* < 0.05; Figure 2h,i). Thus, mice fed the LD had a higher caloric intake and higher rectal temperatures than mice fed the CD, whereas mice fed the FOD showed similar caloric intake to mice fed the CD but higher rectal temperatures.

### 2.2. High-Fat Diets Increase the Expression of Heat-Producing Genes in Brown Adipose Tissue but Not in White Adipose Tissue

To confirm the mechanism by which rectal temperature increased in mice fed the LD and the FOD, mRNA expression of genes involved in heat production was measured in the adipose tissue of mice from each group (Figure 3 and Figure 4).

In BAT, mRNA expression of β_3_AR (*Adrb3*) was 2.5- and 3.7-fold higher than mice fed the CD in mice fed the LD and the FOD, respectively (both *p* < 0.05; Figure 3a). Expression of *Prdm16*, *Ppargc1a*, *Ppara*, and *Pparg* was also significantly higher in mice fed the LD and the FOD than in mice fed the CD in BAT (all *p* < 0.05; Figure 3b–e). Expression of *Ucp1* in BAT was also significantly higher in mice fed both the LD and the FOD than the CD (both *p* < 0.01; Figure 3f) but tended to be lower in mice fed the LD and the FOD than those fed the CD in inguinal WAT (*p* = 0.12; Figure 4f). The expression of the free fatty acid receptor *Ffar4* was significantly higher in mice fed the LD and the FOD compared with mice fed the CD in BAT (both *p* < 0.05; Figure 3i). Expression of *Fgf21* was significantly higher in mice fed the LD and the FOD than the CD (both *p* < 0.05; Figure 3g), showing a different trend from that of plasma FGF21 levels, where mice fed the FOD had significantly lower levels than mice fed the LD (*p* < 0.05; Table 1). βklotho (*Klb*), which acts as a receptor for FGF21, was significantly higher in mice fed the FOD than in mice fed the LD in BAT (Figure 3h). Unlike the results related to BAT, no significant changes in gene expression were observed in inguinal WAT between groups (Figure 4a–i).

### 2.3. Fish Oil Improves Glucose Tolerance and Insulin Sensitivity in Mice with Diet-Induced Obesity

DIO mice fed the FOD had better glucose tolerance than mice fed the LD according to the intraperitoneal GTT (Figure 5a); the area under the curve of blood glucose levels in mice fed the FOD was significantly lower than in mice fed the LD (*p* < 0.01; Figure 5b). Plasma insulin levels were significantly lower in mice fed the FOD both before and after glucose administration than in mice fed the LD (*p* < 0.05; Figure 5c); the area under the curve of blood glucose levels in mice fed the FOD also tended to be lower than in mice fed the LD (*p* = 0.065; Figure 5d). This suggests that insulin resistance was improved by administering the FOD to DIO mice (Figure 5a–d). Intraperitoneal ITTs showed that mice fed the FOD had better insulin sensitivity than mice fed the LD: mice fed the FOD had significantly lower blood glucose levels than mice fed the LD, both before and after insulin administration (*p* < 0.01; Figure 5e); in addition, the area under the curve of blood glucose levels in mice fed the FOD was significantly lower than in mice fed the LD (*p* < 0.01; Figure 5f).

### 2.4. Fish Oil Improves Fasting Blood Glucose, Insulin, TC, FGF21, and Leptin Concentrations in Mice with Diet-Induced Obesity

Plasma TC concentrations in mice fed the FOD were comparable to those in mice fed the CD but significantly lower than in mice fed the LD (*p* < 0.05; Table 1). Mice fed the LD had significantly higher TC levels than mice fed the CD (*p* < 0.01; Table 1). Plasma TG concentrations did not differ between groups. NEFA concentrations tended to be lower in mice fed the FOD than in mice fed the LD, but the difference was not significant. Blood FGF21 concentrations were significantly lower in mice fed the FOD than in mice fed the LD (*p* < 0.05; Table 1). Compared with mice fed the LD, blood leptin levels in mice fed with both the CD and the FOD were significantly lower (both *p* < 0.001), which suggests leptin resistance in mice fed the LD but not in mice fed the FOD (Table 1).

## 3. Discussion

In this study, we found two important results. First, a FOD prevented high-fat diet-induced weight gain by suppressing food intake and maintaining rectal temperatures in mice. Second, high-fat diets stimulated heat-generating gene expression in BAT but not in WAT. 

In the present study, the FOD prevented body weight gain induced by a high-fat diet in mice through both a reduction in food intake and the maintenance of rectal temperature. A transient decrease in energy intake in the FOD group was observed after changing the diet from the LD to the FOD at 13 weeks of age. It has been reported that a lard-based high-fat diet is highly palatable, and that a transient decrease in energy intake occurs after changing from a lard-based high-fat diet to another diet [15,16]. The hypothalamic neurons that express the Agouti-related peptide (AgRP) and the pro-opiomelanocortin (POMC) neurotransmitters have been shown to regulate food intake in a reciprocal manner [17]. There are multiple reports that the changes in mRNA expression of neuropeptide Y (*Npy*), *Pomc*, and *Agrp* caused by a lard-based high-fat diet were improved by the administration of fish oil, although the results vary in terms of the direction of the change in expression of neuropeptide [18,19,20]. We measured the mRNA expressions of *Npy*, *Pomc*, and *Agrp* mRNA in the hypothalamus of mice under non-fasting conditions at 21 weeks of age. There were no significant differences in the expression of these genes among the CD, LD, and FOD groups (Appendix A). At least in our experimental condition, the effects of fish oil on the satiety center may not be direct. Change in food preferences may reduce weight gain in the FOD group, as previous reports showed by direct comparison in mice [21].

The FOD resulted in a rectal temperature similar to the LD, suggesting that mice fed the FOD burned more energy than those fed the CD (Figure 2h,i). DIO mice have increased heat production in BAT due to the hyperleptinemia associated with obesity [13,22]. Leptin is produced and secreted by adipocytes and is responsible for energy homeostasis [22,23]. Although plasma leptin levels are higher in obese animals, these animals are resistant to leptin’s effect on appetite control. However, they are not resistant to its centrally mediated effects on sympathetic activation [22,24]. It is these effects that cause obesity-induced thermogenesis: in BAT and inguinal WAT, β_3_AR signaling induces PRDM16 and PGC1α via the β_3_AR–cAMP–PKA pathway and increases UCP1 expression in mice [11,25,26]; PPARα and PPARγ are also involved in heat production through this signaling pathway [25,27,28]. The absence of change in the expression levels of *Ppargc1a*, *Ppara*, and *Pparg* in the WAT of the LD and FOD groups in this study may be attributable to the unaltered expression levels of *Adrb3*. It is known that the expression of the *Ppargc1a* gene is reduced in the WAT of mice fed a LD [29,30,31], but there are also multiple reports that there is no difference from mice fed a standard diet [32,33,34]. The reason for this discrepancy in results is unclear, but it may be due to differences in the composition or administration period of the lard-based high-fat diet.

In non-obese mice, eicosapentaenoic acid and docosahexaenoic acid activates the sympathetic nervous system via the activation of TRPV1 expressing on the afferent nerves in the gastrointestinal tract, leading to the upregulation of UCP1 expression in BAT and inguinal WAT in a leptin-independent manner [6,35]. Consequently, in non-obese mice, FO increases β_3_AR expression in BAT and WAT [6], thereby increasing UCP1 expression and heat production in BAT and WAT [7,8,9]. Since the induction of *Ucp1* mRNA expression in BAT and WAT by FO is abolished by beta-blocker administration [6], beta-receptor stimulation appears to play an important role in FO-induced heat production in adipose tissue. In the present study, we found that increased expression of genes involved in heat production in BAT occurred in DIO mice fed both the LD and the FOD. Although elevated *Ucp1* mRNA expression does not definitively confirm heat generation in BAT, elevated rectal temperatures further suggest that heat production was activated. These results suggest that enhanced sympathetic signaling was involved in the increased rectal temperatures in mice fed both the LD and the FOD, but the mechanism of sympathetic activation in the two groups was likely different.

We found different heat-generating gene expression profiles between BAT and WAT. Feeding FO to DIO mice in the present study increased *Fgf21* expression in BAT but not in inguinal WAT. FGF21 contributes to heat production by inducing UCP1 expression in adipose tissue [36,37], and FO feeding increases FGF21 expression in both the BAT and WAT of nonobese mice [7]. Since β_3_AR signaling increases *Fgf21* mRNA expression in BAT without an increase in blood FGF21 levels [38,39], there may be a link between the observation that FO feeding increased both *Adrb3* and *Fgf21* mRNA expression only in the BAT and not in the WAT of DIO mice. The fatty acid receptor FFAR4 is highly expressed in adipose tissue [40], and stimulation of FFAR4 also increases FGF21 expression in adipose tissue via the p38 MAPK signaling pathway [41]. In our experiments, the expression of *Ardb3* and *Ffar4* in mice fed the LD and the FOD increased only in BAT and not in inguinal WAT, which may be the reason that *Fgf21* expression increased only in BAT and not in WAT.

Unlike the results of *Fgf21* mRNA expression in BAT, we found that blood FGF21 levels were lower in mice fed the FOD than in mice fed the LD. Circulating FGF21 is mainly derived from liver production [42,43,44], while adipocyte-derived FGF21 acts locally in an autocrine/paracrine manner [37,45]. Therefore, it is unlikely that the induction of *Fgf21* expression in BAT affected blood FGF21 levels. In our study, blood FGF21 levels in mice fed the FOD were lower than those in mice fed the LD, and *Klb* mRNA expression in BAT was higher in mice fed the FOD than in mice fed the LD, which suggests an increased sensitivity to FGF21 in BAT. Previous reports show that the expression of βKlotho, a coreceptor of FGF21, is decreased in the obese state [46,47] and that blood FGF21 concentration is increased due to FGF21 resistance [48,49]. Although we have not found any reports mentioning FO-induced FGF21 sensitivity in adipose tissue, there are suggestions of FO-induced increases in FGF21 sensitivity resulting from increased expression of βKlotho in the liver [47,50]. Thus, induction of FGF21 expression and increased sensitivity to FGF21 may occur in BAT by feeding FO to DIO mice; however, the details and significance of this phenomenon should be verified by future studies.

The present study is novel in that it examined the expression of genes involved in heat production in both BAT and WAT after feeding FO to obese mice. However, this study showed that FO is unlikely to have a beiging effect on WAT in obesity, as demonstrated by a lack of change in heat-generating genes in WAT. There are several limitations of the present study. First, because we did not measure energy metabolism using a metabolic cage, we cannot confirm whether there is a strict relationship between increased rectal temperature and increased expression of heat-producing genes in brown adipocytes. It is also unclear to what extent the rise in rectal temperature in the FOD group had an impact on weight loss. Second, we did not compare FO-fed obese and non-obese mice, meaning we could not confirm whether the observed effects of FO are specific to an obese state. Third, these experimental results in mice cannot necessarily be applied to humans. In humans, the content and activity of BAT decreases with age [51]. Therefore, the anti-obesity effect of FO in obese individuals via heat production may diminish with age. Interestingly, meta-analyses of randomized controlled trials examining the weight-loss effects of FO in obese humans do not consistently demonstrate an associated weight-loss effect [3,52], although several studies in relatively young obese individuals (<40 years old) do show a weight-loss effect with FO [53,54]. Therefore, further research is needed to confirm the anti-obesity effect of FO via heat generation in humans.

## 4. Materials and Methods

### 4.1. Animals and Diets

Four-week-old male C57BL/6J mice were purchased from KBT Oriental Co., Ltd. (Saga, Japan) and housed at 23 ± 1 °C on a 12-h light/12-h dark cycle with ad libitum access to food and water. After a 1-week acclimation period, mice were divided into two groups: one was fed a control diet (*n* = 6) (CD) (D20051305M, Research Diets Inc., New Brunswick, NJ, USA) and the other was fed a lard-based high-fat diet (*n* = 16) (LD) (D20051306M) for 8 weeks. The mice that were fed the LD for 8 weeks were randomly assigned to one of the following regimens: continuation of an LD for another 8 weeks (*n* = 8) or introduction to a fish oil-based high-fat diet (*n* = 8) (FOD) (D20051307M) for another 8 weeks. The mice fed the CD were maintained on this diet for another 8 weeks. The CD contained 9% calories from fat, 21% calories from protein, and 70% calories from carbohydrates (3.8 kcal/g); the LD and the FOD contained 60% calories from fat, 21% calories from protein, and 19% calories from carbohydrates (5.3 kcal/g) (Table 2). The LD contained 67.3g of n-6 PUFA and 4.5 g of n-3 PUFA in 1 kg of diet; the FOD contained 18.8 g and 77.6 g, respectively (Appendix A). Food was provided to mice every other day. To estimate daily food intake, the food weight of each day was subtracted from the initial food weight of the previous day. The mean calorie intake in each of the three groups was calculated using these data. Mice were weighed after a 6-h fast at the end of each week. With the exception of mice at 20 weeks of age, rectal temperatures were measured weekly under fed conditions at 7:00 a.m., which represented the end of the dark period with reference to past reports [6,55]. Because glucose and insulin tolerance tests (GTTs and ITTs, respectively) were performed at 20 weeks of age, rectal temperature was measured in the state of overnight fast only at 20 weeks of age. At 21 weeks of age, the mice were sacrificed. All experimental procedures were reviewed and approved by the Laboratory Animal Committees of Kagoshima University Graduate School and were performed in accordance with the guidelines for the care and use of laboratory animals and the ARRIVE guidelines.

### 4.2. Measurement of Metabolic Parameters

Blood glucose was obtained from the mice, and plasma levels of nonesterified fatty acids (NEFAs), triglycerides (TGs), and total cholesterol (TC) were measured by enzymatic colorimetry with LabAssay kits (Fujifilm Wako Pure Chemical Corporation, Osaka, Japan). Plasma fibroblast growth factor 21 (FGF21) was measured using a Mouse/Rat FGF21 Quantikine ELISA Kit (R&D Systems Inc., Minneapolis, MN, USA). Plasma leptin was measured using a Mouse Leptin ELISA Kit (Proteintech Group Inc., Rosemont, IL, USA). Plasma insulin was measured using a mouse insulin ELISA kit (Morinaga Institute of Biological Science, Inc., Kanagawa, Japan). TGs and TC were measured under fasting conditions. Because rectal temperature measurements were performed under nonfasting conditions, NEFAs, FGF21, and leptin were also measured under nonfasting conditions to investigate their relationship with body temperature.

### 4.3. Glucose and Insulin Tolerance Tests

The GTT was performed in the morning after an overnight fast. D-Glucose (2 mg/g body weight) was injected intraperitoneally. Capillary blood samples were collected using the tail cut method, and blood glucose was measured with Stat Strip XP3 (NIPRO Corporation, Osaka, Japan) before glucose injection and at 30, 60, and 120 min after glucose injection. The ITT was performed in the morning after an overnight fast. Insulin (Humulin R, Eli Lilly Japan K.K., Hyogo, Japan) was injected intraperitoneally at 0.1 mU/g body weight, and blood glucose was measured before insulin injection and at 30, 60, and 120 min after insulin injection.

### 4.4. Histology

Twenty-one-week-old mice were intraperitoneally administered 90 mg/kg pentobarbital after inhaling anesthesia with isoflurane. The mice were systemically perfused with phosphate-buffered saline from the left ventricle and then perfused with 4% paraformaldehyde and fixed. After fixation, livers were removed and Oil red O staining was performed at the Bio-Pathology Laboratory (Oita, Japan).

### 4.5. Quantitative Real-Time PCR and Organ Weights

Mice were anesthetized with pentobarbital 90 mg/kg intraperitoneally at 21 weeks of age, and inguinal adipose, brown adipose, epididymal adipose, mesenteric adipose, and liver tissues were harvested after general perfusion with phosphate-buffered saline. RNA was extracted from tissues with TRIzol Reagent (Life Technologies Japan Ltd., Tokyo, Japan) following the manufacturer’s instructions. Reverse transcription was performed using High Capacity cDNA Reverse Transcription Kits (Thermo Fisher Scientific K.K., Tokyo, Japan). Quantitative real-time PCR was performed on a StepOnePlus Real-Time PCR system with TaqMan Fast Universal PCR Master Mix (Thermo Fisher Scientific K.K.) and the following gene-specific primers: *Gapdh*, Mm99999915_g1; *Ucp1*, Mm01244861_m1; *Prdm16*, Mm00712556_m1; *Fgf21*, Mm00840165_g1; *Klb (βklotho)*, Mm00502002_m1; *Adrb3*, Mm02601819_g1; *Ppargc1a*, Mm01208835_m1; *Ppara*, Mm00440939_m1; *Pparg*, Mm01184322_m1; and *Ffar4* Mm00725193_m1. Relative gene expression was calculated using the ΔΔCt method. The expression of target genes was normalized to that of *Gapdh*.

### 4.6. Statistical Analyses

All numerical values are expressed as the mean ± standard error of the mean. A two-sample *t*-test was used for comparisons between two groups. Comparisons between multiple groups were performed with a one-way analysis of variance (ANOVA), and, based on results of Levene’s test, multiple comparisons were performed using Tukey’s or Games–Howell post-hoc analysis. Statistical significance was indicated by a *p*-value < 0.05. All statistical analyses were performed using the Statistical Package for the Social Sciences (SPSS) 29.0 (IBM Corp., Armonk, NY, USA).

## 5. Conclusions

In conclusion, we showed that FO increases rectal temperature and exerts obesity-suppressing effects in DIO mice and that high-fat diets increase the expression of heat-producing genes in BAT but not WAT. Fish oil may therefore be a potential therapeutic approach to obesity.

## Figures and Tables

**Figure 1 ijms-26-00302-f001:**
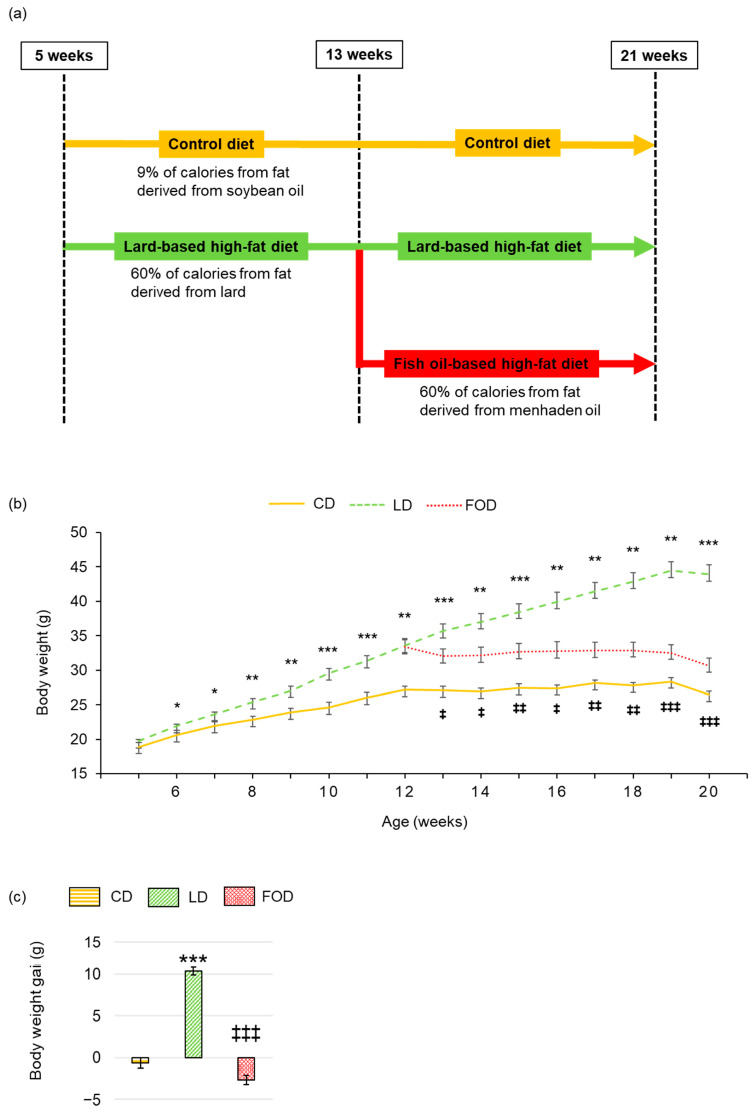
Fish oil prevents dietary fat-induced increases in body weight in mice with diet-induced obesity. (**a**) Experimental protocol: Five-week-old male C57BL/6J mice were fed a control diet (CD) or a lard-based high-fat diet (LD) for 8 weeks. (**b**) Body weight; (**c**) body weight gain between 12 and 20 weeks of age. The CD contained 9% calories from fat, 21% calories from protein, and 70% calories from carbohydrates (3.8 kcal/g); the LD and a fish oil-based high-fat diet (FOD) contained 60% calories from fat, 21% calories from protein, and 19% calories from carbohydrates (5.3 kcal/g). Food was provided to the mice every other day. The data are presented as the mean ± standard error of the mean, *n* = 4–6 animals per group * *p* < 0.05, ** *p* < 0.01, *** *p* < 0.001 compared with the CD group, ‡ *p* < 0.05, ‡‡ *p* < 0.01, ‡‡‡ *p* < 0.001 compared with the LD group using one-way ANOVA with Tukey’s or Games–Howell multiple comparisons post hoc test based on Levene’s test results.

**Figure 2 ijms-26-00302-f002:**
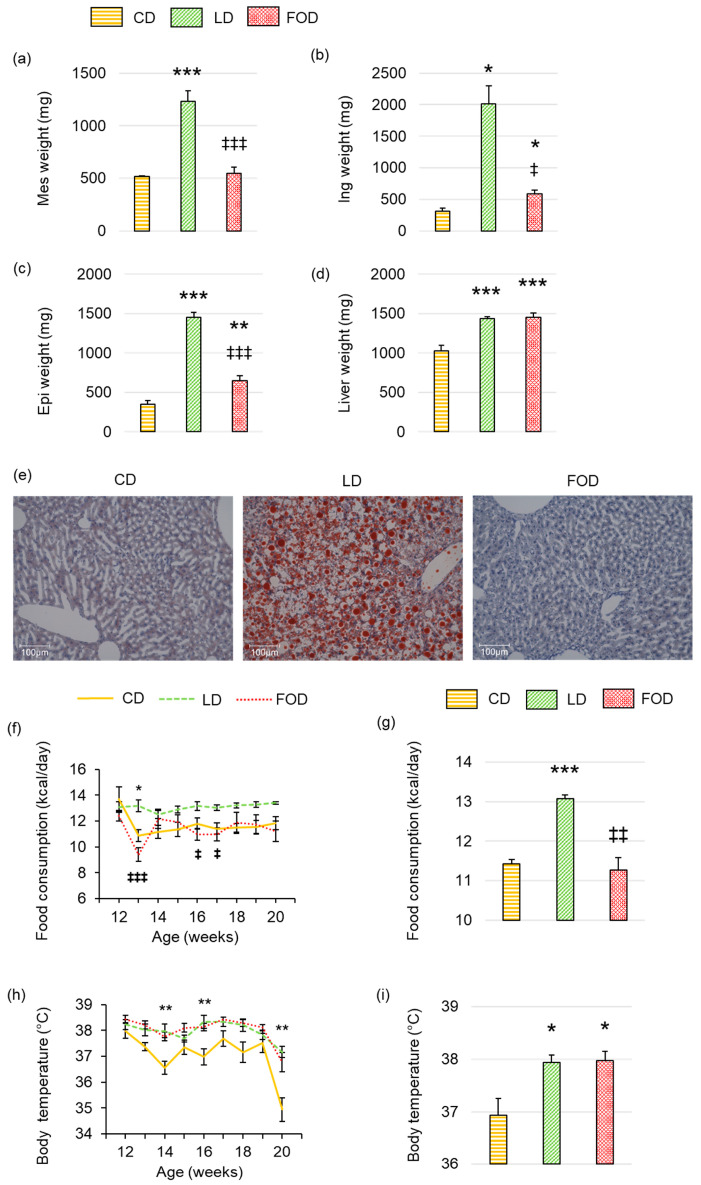
Fish oil prevents dietary fat-induced increases in adipose tissue mass and food intake while maintaining rectal temperature in mice with diet-induced obesity. (**a**) Mesenteric adipose tissue (Mes) mass; (**b**) inguinal adipose tissue (Ing) mass; (**c**) epididymal adipose tissue (Epi) mass; (**d**) liver mass; (**e**) representative Oil red O-stained liver histology; (**f**) food intake; (**g**) average food intake between 13 and 20 weeks of age; (**h**) body temperature; (**i**) average body temperature between 13 and 20 weeks of age. The data are presented as the mean ± standard error of the mean, *n* = 4–6 animals per group. * *p* < 0.05, ** *p* < 0.01, *** *p* < 0.001 compared with the CD group; ‡ *p* < 0.05, ‡‡ *p* < 0.01, ‡‡‡ *p* < 0.001 compared with the LD group using a one-way ANOVA with Tukey’s or Games–Howell multiple comparisons post hoc test based on Levene’s test results. DIO, diet-induced obesity; LD, lard-based high-fat diet; CD, control diet; FOD, fish oil-based high-fat diet.

**Figure 3 ijms-26-00302-f003:**
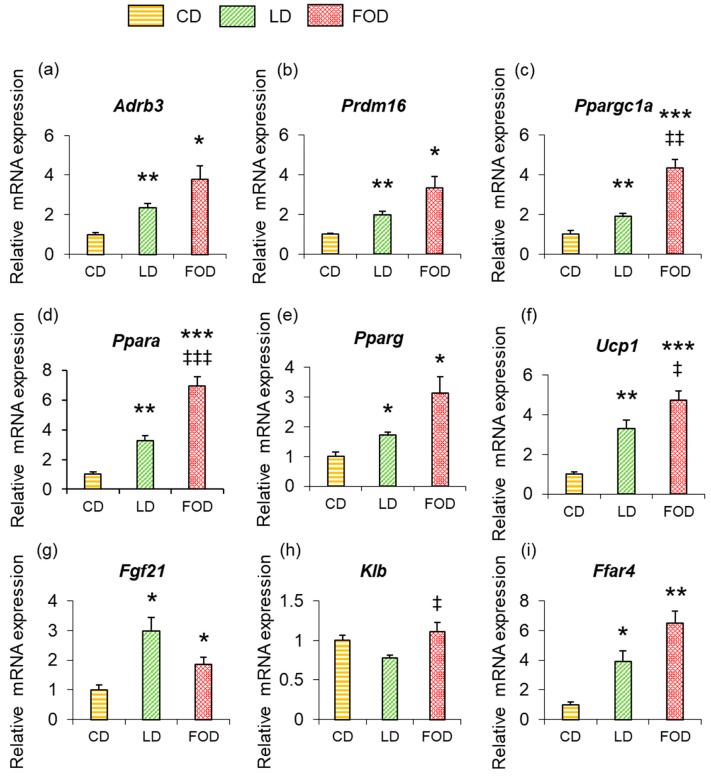
High-fat diets increase the expression of heat-producing genes in brown adipose tissue (BAT). The expression of genes involved in heat production increased in the BAT of mice fed the LD and the FOD. Gene expression levels in BAT. (**a**) Adrb3; (**b**) Prdm16; (**c**) *Ppargc1a*; (**d**) *Ppara*; (**e**) *Pparg*; (**f**) *Ucp1*; (**g**) *Fgf21*; (**h**) *Klb*; (**i**) *Ffar4*. The data are presented as the mean ± standard error of the mean, *n* = 6 animals per group. * *p* < 0.05, ** *p* < 0.01, *** *p* < 0.001 compared with the CD group, ‡ *p* < 0.05, ‡‡ *p* < 0.01, ‡‡‡ *p* < 0.001 compared with the LD group using a one-way ANOVA with Tukey’s or Games–Howell multiple comparisons post hoc test based on Levene’s test results. BAT, brown adipose tissue; CD, control diet; LD, lard-based high-fat diet; FOD, fish-oil based high-fat diet; *Adrb3*, β_3_-adrenergic receptor; *Prdm16*, PR domain containing 16; *Ppargc1a*, peroxisome proliferator-activated receptor-γ coactivator-1α; *Ppara*, peroxisome proliferator-activated receptor-α; *Pparg*, peroxisome proliferator-activated receptor-γ; *Ucp1*, uncoupling protein 1; *Fgf21*, fibroblast growth factor 21; *Klb*, βklotho; *Ffar4*, free fatty acid receptor 4.

**Figure 4 ijms-26-00302-f004:**
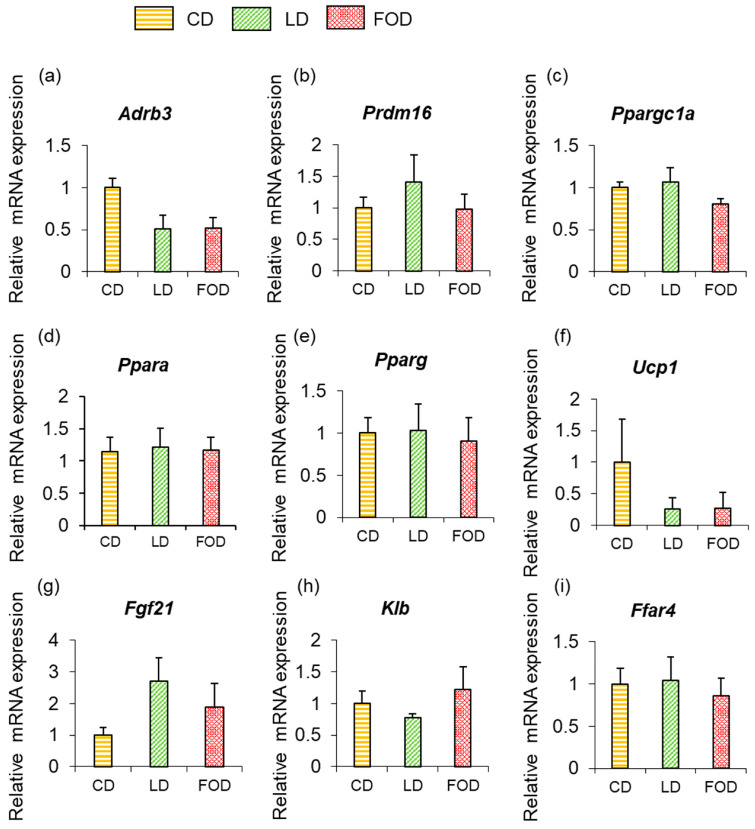
High-fat diets do not increase the expression of heat-producing genes in inguinal white adipose tissue (WAT). No changes in gene expression were observed in inguinal WAT. (**a**) *Adrb3*; (**b**) *Prdm16*; (**c**) *Ppargc1a*; (**d**) *Ppara*; (**e**) *Pparg*; (**f**) *Ucp1*; (**g**) *Fgf21*; (**h**) *Klb*; (**i**) Ffar4. The data are presented as the mean ± standard error of the mean, *n* = 6 animals per group. WAT, white adipose tissue; CD, control diet; LD, lard-based high-fat diet; FOD, fish-oil based high-fat diet; *Adrb3*, β_3_-adrenergic receptor; *Prdm16*, PR domain containing 16; *Ppargc1a*, peroxisome proliferator-activated receptor-γ coactivator-1α; *Ppara*, peroxisome proliferator-activated receptor-α; *Pparg*, peroxisome proliferator-activated receptor-γ; *Ucp1*, uncoupling protein 1; *Fgf21*, fibroblast growth factor 21; *Klb*, βklotho; *Ffar4*, free fatty acid receptor 4.

**Figure 5 ijms-26-00302-f005:**
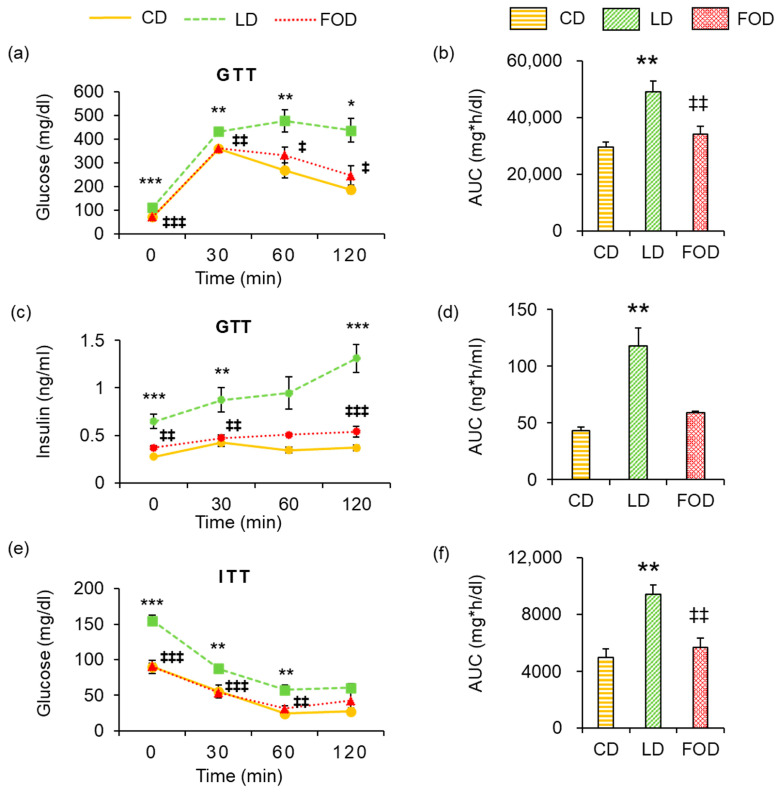
Fish oil improves glucose tolerance and insulin resistance in mice with diet-induced obesity. Mice were dissected at 21 weeks. (**a**) Changes in blood glucose, as indicated by the GTT. (**b**) AUC of blood glucose levels during the GTT. (**c**) Changes in plasma insulin, as indicated by the GTT. (**d**) AUC of plasma insulin levels during the GTT. (**e**) Changes in blood glucose, as indicated by the ITT. (**f**) AUC of blood glucose levels during the ITT. The data are presented as the mean ± standard error of the mean, *n* = 4–8 animals per group. * *p* < 0.05, ** *p* < 0.01, *** *p* < 0.001 compared with the CD group, ‡ *p* < 0.05, ‡‡ *p* < 0.01, ‡‡‡ *p* <0.001 compared with the LD group using a one-way ANOVA with Tukey’s or Games–Howell multiple comparisons post hoc test based on Levene’s test results. CD, control diet; LD, lard-based high-fat diet; FOD, fish oil-based high-fat diet; ITT, insulin tolerance test; AUC, area under the curve; GTT, glucose tolerance test.

**Table 1 ijms-26-00302-t001:** Plasma concentrations of biochemical parameters in 21-week-old male C57BL/6J mice.

	CD	LD	FOD
FBG (mg/dL)	71.2 ± 5.0	110.8 ± 5.3 ***	70.2 ± 4.0 ^‡‡‡^
Insulin (ng/mL)	0.3 ± 0	0.6 ± 0.1 *	0.4 ± 0 *
TC (mg/dL)	45.1 ± 11.7	68.7 ± 7.0 **	52.2 ± 8.3 ^‡^
TG (mg/dL)	69.7 ± 24.0	89.8 ± 11.4	90.7 ± 22.2
NEFAs (mEq/L)	0.1 ± 0	0.2 ± 0	0.1 ± 0
FGF21 (pg/mL)	1146.4 ± 375.2	1276.1 ± 188.4	282.7 ± 44.6 ^‡^
Leptin (ng/mL)	14.4 ± 3.3	42.3 ± 3.7 ***	16.9 ± 3.7 ^‡‡‡^

The data are presented as the mean ± standard error of the mean, *n* = 4–6 animals per group. The second decimal place and below have been rounded off. * *p* < 0.05, ** *p* < 0.01, *** *p* < 0.001 compared with the CD group, ‡ *p* < 0.05, ‡‡‡ *p* < 0.001 compared with the LD group using a one-way ANOVA with Tukey’s or Games-Howell multiple comparisons post hoc test based on Levene’s test results. FBG, fasting blood glucose; TC, total cholesterol; TG, triglyceride; NEFAs, nonesterified fatty acids; FGF21, fibroblast growth factor 21; CD, control diet; LD, lard-based high-fat diet; FOD, fish oil-based high-fat diet.

**Table 2 ijms-26-00302-t002:** Composition of experimental diets.

	CD	LD	FOD
g/kg			
Casein	200	200	200
L-Cystine	3	3	3
Corn Starch	483.5	0	0
Maltodextrin 10	75	75	75
Sucrose	101	101	101
Cellulose	50	50	50
Soybean Oil	40	15	15
Menhaden Oil	0	0	240
Lard	0	240	0
BHA	0.017	(0.017) *	0.017
BHT	0.012	(0.012) *	0.012
Mineral Mix	35	35	35
Vitamin Mix	10	10	10
Choline Bitartrate	2.5	2.5	2.5
Kcal (%)			
Protein	21	21	21
Carbohydrate	70	19	19
Fat	9	60	60
Kcal/g	3.8	5.3	5.3

* 240 g lard contains 0.017 g BHA and 0.012 g BHT. CD, control diet; LD, lard-based high-fat diet; FOD, fish oil-based high-fat diet; BHA, butylated hydroxyanisole; BHT, butylated hydroxytoluene.

## Data Availability

Data is contained within the article and Appendix A.

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
