# Peer review of "The Anti-Obesity Effect of Fish Oil in Diet-Induced Obese Mice Occurs via Both Decreased Food Intake and the Induction of Heat Production Genes in Brown but Not White Adipose Tissue"

_ijms, 2024, doi:10.3390/ijms26010302_

Round 1

Reviewer 1 Report

Comments and Suggestions for Authors

In the manuscript ijms-3360606, the authors have demonstrated that fish oil supplementation reversed lard-induced obesity by inducing brown fat thermogenesis. This study has a strength in measuring body temperature and multiple indicators in serum, liver, and multiple depots of adipose tissue; however, there are some concerns and issues required to be addressed.

Major concerns

1.       Could the authors provide fatty acid composition of the experimental diets or menhaden oil? Without the composition, it is hard to mention that the anti-obesity effects of FOD are due to n-3 fatty acids.

2.       It is interesting that FOD group had less energy intake (Figure 2F), but there was no explanation about that. I wonder if neuropeptide expressions in hypothalamus differ.

3.       It seems weird that CD mice had very low rectal temperature at week 20 (Figure 2H). Could the authors explain why?

Minor issues

1.       In Figure 2A-D, it seems better to show the tissue weight as percentage over body weight.

2.       If the authors have BAT and muscle weight data, please show them in Figure 2.

3.       In Figure 5A-C, please show statistical differences at every time point.

4.       To increase the readability, please show all the data with one decimal place in table 1.

5.       There is no reference 42 cited in the manuscript.

Author Response

To Reviewer 1

We thank reviewer 1 for the positive evaluation to our manuscript.

Major concerns

Comments 1: Could the authors provide fatty acid composition of the experimental diets or menhaden oil? Without the composition, it is hard to mention that the anti-obesity effects of FOD are due to n-3 fatty acids.

Response: Thank you for pointing this out. The fatty acid composition of the diet used in the experiment has been added as Supplementary Table S1 (Line 309).

Comments 2: It is interesting that FOD group had less energy intake (Figure 2F), but there was no explanation about that. I wonder if neuropeptide expressions in hypothalamus differ.

Response: Thank you for your important remarks. Indeed, we have explored the expressions of neuropeptide in hypothalamus but we do not have a clear answer, unfortunately. We used quantitative real-time PCR to investigate the expression of neuropeptide Y (Npy), pro-opiomelanocortin (Pomc), and agouti-related peptide (Agrp) mRNA in the hypothalamus of mice under non-fasting conditions at 21 weeks of age. However, there were no significant differences in the expression of these genes among the CD, LD, and FOD groups (the results have been added as Supplementary Figure S1 in the revised version of our manuscript). In previous reports, there are multiple reports that the changes in mRNA expression of Npy, Pomc, and Agrp caused by a lard-based high-fat diet were improved by the administration of ω3 PUFA, although the results vary in terms of the direction of the change in expression of neuropeptide [1][2][3]. We have added this in the Discussion section (Line 283-292).

[1] Wang H, Storlien LH, Huang XF. Effects of dietary fat types on body fatness, leptin, and ARC leptin receptor, NPY, and AgRP mRNA expression. Am J Physiol Endocrinol Metab. 2002 Jun;282(6):E1352-9. doi: 10.1152/ajpendo.00230.2001. PMID: 12006366.

[2] Huang XF, Xin X, McLennan P, Storlien L. Role of fat amount and type in ameliorating diet-induced obesity: insights at the level of hypothalamic arcuate nucleus leptin receptor, neuropeptide Y and pro-opiomelanocortin mRNA expression. Diabetes Obes Metab. 2004 Jan;6(1):35-44. doi: 10.1111/j.1463-1326.2004.00312.x. PMID: 14686961.

[3] Gout J, Sarafian D, Tirard J, Blondet A, Vigier M, Rajas F, Mithieux G, Begeot M, Naville D. Leptin infusion and obesity in mouse cause alterations in the hypothalamic melanocortin system. Obesity (Silver Spring). 2008 Aug;16(8):1763-9. doi: 10.1038/oby.2008.303. Epub 2008 Jun 12. PMID: 18551122.

Comments 3:  It seems weird that CD mice had very low rectal temperature at week 20 (Figure 2H). Could the authors explain why?

Response: Thank you for your very important comment. Rectal temperatures were measured at 7:00 am every week in a non-fasting state. However, because glucose and insulin tolerance tests were conducted at 20 weeks of age, rectal temperatures were measured in overnight fasting only at 20 weeks of age. This is thought to have caused a decrease in rectal temperatures in all groups. As this was not sufficiently described, we have added this to the methods section (Line 313-314, 316-318).

Minor issues

Comments 1: In Figure 2A-D, it seems better to show the tissue weight as percentage over body weight.

Response: We made Figure 2A-D with % body weight as suggested (Figures for Reviewer only). Most of the results were similar to the original version, absolute value, however, liver wight by % body weight were the smallest in LD group probably because increased WAT mass. Thus, we prefer to use the absolute value.

Comments 2: If the authors have BAT and muscle weight data, please show them in Figure 2.

Response: We do not measure BAT and muscle mass. We cannot show these results.

Comments 3: In Figure 5A-C, please show statistical differences at every time point.

Response: As suggested, statistical differences have been added to Figures 5a, c, and e.

Comments 4: To increase the readability, please show all the data with one decimal place in table 1.

Response: The data in Table 1 has been rounded off to the nearest hundredth, accordingly.

Comments 5:  There is no reference 42 cited in the manuscript.

Response: It was quoted on line 302 in the Method section (Line 316).

Reviewer 2 Report

Comments and Suggestions for Authors

The manuscript title was “The anti-obesity effect of fish oil in diet-induced obese mice occurs via both decreased food intake and the induction of heat production genes in brown but not white adipose tissue”. The research was about the omega-3 (ω-3) polyunsaturated fatty acids in fish oil promote heat production in adipose tissue and prevent diet-induced obesity.The research was meaningful. The specific comments was as follow.

1. The introduction should be summarized the mechanism of anti-obesity, the purpose and significance of this study.

2. The p in (p < 0.001) should be italics.

3. The conclusion should be show the key results and finding in your research.

4. The part of discussion should be move to the results, this maybe more readability.

5. Line 157: The sentence This indicates that insulin resistance was improved by ad- 157

ministering the FOD to DIO mice.  The reference should be given.

6. Line 121: The Fig. 3g was Figure. 3g.

Author Response

To Reviewer 2

We thank reviewer 2 for the positive evaluation to our manuscript.

Comments 1: The introduction should be summarized the mechanism of anti-obesity, the purpose and significance of this study.

Response: In the Introduction, we referred to past reports and indicated the mechanisms that are thought to be involved in the anti-obesity effects of fish oil (lines 45-53), but the purpose and significance of this study were not adequately described. Therefore, we have added a description of the purpose and significance of this study (Line 57-59).

Comments 2: The p in “(p < 0.001)” should be italics.

Response: As suggested, we have made the changes accordingly.

Comments 3: The conclusion should be show the key results and finding in your research.

Response: We had the conclusion section at Line 379-381.  

Comments 4: The part of discussion should be move to the results, this maybe more readability.

Response: Thank you for valuable comment. We discussed with co-authors and decided to keep as it is.

Comments 5: Line 157: The sentence “This indicates that insulin resistance was improved by administering the FOD to DIO mice. ”.  The reference should be given.

Response: In the glucose tolerance test, the FOD group had higher plasma insulin levels than the LD group at all time points before and after glucose administration, and the blood glucose levels were also consistently higher than in the LD group. Therefore, we determined that the FOD group had lower insulin resistance than the LD group. We revised Line 159 to “This suggests that insulin resistance was improved by administering the FOD to DIO mice (Figure 5a-d)”.

Comments 6: Line 121: The “Fig. 3g” was “Figure. 3g”.

Response: Thank you for pointing this out. We have made the changes, as suggested.

Round 2

Reviewer 1 Report

Comments and Suggestions for Authors

The authors addressed all the issues and the issues were applied to the revised manuscript. The manuscript seems to get ready to be published.

Author Response

To Reviewer 1

We thank reviewer 1 for the positive evaluation to our manuscript.

Reviewer 2 Report

Comments and Suggestions for Authors

None

Author Response

To Reviewer 2

We thank reviewer 2 for the positive evaluation to our manuscript.